# Study of Al$_2$O$_3$ Sol-Gel Coatings on X20Cr13 in Artificial North German Basin Geothermal Water at 150 °C

**Gabriela Aristia** , **Le Quynh Hoa** , **Marianne Nofz, Regine Sojref and Ralph Bäßler** *

Bundesanstalt für Materialforschung und-Prüfung (BAM), Unter den Eichen 87, 12205 Berlin, Germany;
gabriela.aristia@bam.de (G.A.); quynh-hoa.le@bam.de (L.Q.H.); marianne.nofz@bam.de (M.N.);
regine.sojref@bam.de (R.S.)
* Correspondence: ralph.baessler@bam.de

**Abstract:** Al$_2$O$_3$ has been widely used as a coating in industrial applications due to its excellent chemical and thermal resistance. Considering high temperatures and aggressive mediums exist in geothermal systems, Al$_2$O$_3$ can be a potential coating candidate to protect steels in geothermal applications. In this study, γ-Al$_2$O$_3$ was used as a coating on martensitic steels by applying AlOOH sol followed by a heat treatment at 600 °C. To evaluate the coating application process, one-, two-, and three-layer coatings were tested in the artificial North German Basin (NGB), containing 166 g/L Cl$^-$, at 150 °C and 1 MPa for 168 h. To reveal the stability of the Al$_2$O$_3$ coating in NGB solution, three-layer coatings were used in exposure tests for 24, 168, 672, and 1296 h, followed by surface and cross-section characterization. SEM images show that the Al$_2$O$_3$ coating was stable up to 1296 h of exposure, where the outer layer mostly transformed into boehmite AlOOH with needle-like crystals dominating the surface. Closer analysis of cross-sections showed that the interface between each layer was affected in long-term exposure tests, which caused local delamination after 168 h of exposure. In separate experiments, electrochemical impedance spectroscopy (EIS) was performed at 150 °C to evaluate the changes of coatings within the first 24 h. Results showed that the most significant decrease in the impedance is within 6 h, which can be associated with the electrolyte penetration through the coating, followed by the formation of AlOOH. Here, results of both short-term EIS measurements (up to 24 h) and long-term exposure tests (up to 1296 h) are discussed.

**Keywords:** Al$_2$O$_3$; geothermal; martensitic steels; boehmite





## 1. Introduction

As an effort to decrease global carbon emissions, geothermal energy is widely considered as an alternative to fossil-fuel-based energy [1]. However, its implementation is often hindered by corrosion and scaling in geothermal powerplants, which are two major impediments toward reliable energy production [2]. For example, materials for geothermal pipelines may interact with the transported geothermal fluid, resulting in corrosion processes. The continuous corrosion processes cause the accumulation of corrosion products or scaling on the surface of the pipelines that may degrade the physical integrity of the pipelines and reduce the flow of geothermal fluid. Therefore, corrosion and scaling have to be mitigated for building safe and efficient geothermal powerplants ensuring reliable energy production.

Depending on the depth and locations of geothermal wells, geothermal fluid with different corrosivity levels can be found. These corrosivity levels are mainly based on the total dissolved salts content, pH, and the amount of corrosive gases in the geothermal fluid [3]. Common types of corrosion that can be found in geothermal powerplants include uniform, pitting, crevice, and stress-corrosion cracking [3]. Along with the corrosivity levels of the geothermal fluid, the severity of corrosion is dependent on the types of materials used in the powerplant. For example, in low-chloride-containing environments at neutral or high pH, carbon steel underwent uniform corrosion with a corrosion rate

that is negligible [4]. On the other hand, in high-chloride-containing as well as in low-pH environments, carbon steel underwent severe uniform corrosion with a very high corrosion rate, whereas stainless steel suffered pitting corrosion [5–7]. To provide reliable energy production, the corrosion rate should be kept below 0.3 mm/year for an estimated loss of 6 mm wall thickness over 20 years, and pitting corrosion should be avoided at any cost [5]. Consequently, high-alloyed materials, such as nickel- and titanium-based alloys are often used in highly aggressive geothermal environments [5,8]. However, these alloys are more expensive than carbon or stainless steel because the base metals, Ni and Ti, are more expensive compared to Fe, and their production processes are more complex. To provide affordable alternatives, protective measures on carbon and stainless steels should be developed by either using coatings or inhibitors [9].

In the past 50 years, different coatings for geothermal applications have been developed and tested in laboratories and on sites [9]. Several types of coatings have been tested in both simulated and real geothermal environments, including metallic [10], polymer (Teflon PFA, PTFE, polyurethane, epoxy, and other organic coatings) [9,11,12], and ceramic coatings [13]. Among these coatings, glass, glass-ceramic, and polymer-derived ceramic coatings stand out due to inert chemical properties, high-temperature resistance, and high mechanical properties [9]. Ceramic particles have also been utilized either directly on steels or as reinforcements or fillers in other matrices of coatings for geothermal applications. Some examples of coatings that incorporate ceramic materials include $SiO_2$ [13,14], $TiO_2$ [15], and $ZrO_2$ [15].

Besides the above-mentioned ceramic-based coatings, $Al_2O_3$ is considered to be one of the most promising candidates for corrosion protection in various environments because it has excellent wear resistance, corrosion resistance, and high mechanical strength [16]. Various methods have been used to synthesize and apply $Al_2O_3$ as coatings, such as chemical vapor deposition [17,18], atomic layer deposition [19,20], plasma spraying deposition [21], magnetron sputtering [22,23], and sol-gel [24,25]. Among these methods, sol-gel is one of the simplest and versatile techniques as it can be applied by dipping, spraying, or spinning [26]. As reported by Schulz et al. and Nofz et. al., $Al_2O_3$ sol-gel coatings were able to protect steels in high-temperature flue gas [24,27]. $Al_2O_3$ sol-gel coatings were also proven to be protective in NaCl-containing environments [25,28]. Given that geothermal environments may have a combination of highly saline water and high temperature, the aim of this study is to identify the suitability of $Al_2O_3$ as a corrosion protective coating in artificial geothermal solutions.

Among different alumina polymorphs, $\alpha$-$Al_2O_3$ is the most thermodynamically stable and favorable form of alumina for coating applications due to its high hardness and thermal and chemical stability [29]. However, there is a limited number of substrates suitable for $\alpha$-$Al_2O_3$ coating because the sintering process requires a temperature of above 1000 °C. Therefore, studies on other polymorphs of $Al_2O_3$ with lower sintering temperatures have been actively performed to provide fundamental knowledge on their properties as coating materials. Based on the sintering temperature, the boehmite precursor transforms to $\gamma$-$Al_2O_3$ between 300–600 °C, to $\delta$-$Al_2O_3$ between 700–800 °C, and to $\theta$-$Al_2O_3$ between 900–1000 °C, which can also be formed concurrently depending on the synthesis method [30]. This study focuses on the use of $\gamma$-$Al_2O_3$ formed by sintering at 600 °C, as this polymorph has been proven to be protective in different corrosive environments [25,27]. According to the Pourbaix diagram, $Al_2O_3$ is stable in room temperature aqueous solutions with pH between 4−9 [31]. Thus, a North German Basin (NGB) artificial solution with pH 6 was used as the medium of choice. Besides, NGB is also considered to be one of the most aggressive geothermal waters because of its high chloride concentration. In this study, the $Al_2O_3$ coatings were evaluated by varying the coating thicknesses and exposing the coated specimens to the NGB solution. The stability of $Al_2O_3$ sol-gel coatings was also observed by conducting exposure tests for three months. In addition, the corrosion resistance of coatings was also evaluated and discussed by comparing the results of three-month exposure tests

and short-term electrochemical impedance measurements in the deaerated NGB solution at 150 °C and 1 MPa.

## 2. Materials and Methods

X20Cr13 steels were used as substrates, with the chemical composition analyzed using a spark emission spectrometer (Table 1). For exposure tests, these steel substrates were machined into specimens with dimensions of 27 mm × 10 mm × 5 mm. A hole, with a diameter of 3 mm, was drilled to position the specimens in the autoclave. For electrochemical tests, the dimension was 27 mm × 15 mm × 5 mm. All specimens were ground using SiC abrasive paper, from grades P120, P320, P600, until P1200. Specimens were cleaned using bi-distilled water and ethanol thoroughly and ultrasonicated in ethanol for 5 min prior to the experiments.

**Table 1.** Chemical composition of X20Cr13 (in wt.%) measured by optical emission spectroscopy.

| Fe | Cr | Si | Mn | C | Ni | V | Cu | Mo |
|------|-------|------|------|------|------|-------|------|------|
| 85.7 | 12.95 | 0.45 | 0.39 | 0.23 | 0.12 | 0.067 | 0.03 | 0.03 |
| **Co** | **P** | **Al** | **Ti** | **Sn** | **Pb** | **S** | **Nb** | |
| 0.02 | 0.016 | 0.008 | 0.004 | 0.004 | 0.001 | 0.001 | <0.005 | |

After they were ground with P1200 SiC paper as the last preparation step, the surface roughness of the X20Cr13 specimens was measured based on the microscopic images collected using the laser scanning microscope (LSM, LEXT 4100, Olympus, Hamburg, Germany) at 50 times magnification. The surface roughness average ($R_a$) of X20Cr13 is 0.018 μm, with a maximum sink height ($R_z$) of 0.155 μm.

### 2.1. Coating Application

Coatings were applied to X20Cr13 specimens by the sol-gel method, as described by Nofz, et al. [24]. Sol was prepared using boehmite (Sasol) stirred in ethanol at room temperature for 20 min. Polyvinylbutyral type PVB B76 (Butvar) was dissolved in ethanol and added as a binder. A small amount of isopropanol was also added to improve the solubility of the binder in the solution. The sol was then treated with ultrasound for 5−10 min in a water bath cooled with ice to achieve a good dispersion with the small particle size. The final sol contained 15 wt% boehmite and 45 mg PVB per gram sol. The resulting sol was Newtonian within the shear rate range of 1 and 500 s$^{-1}$, with a viscosity of 20 MPa s.

To apply the alumina coating, the specimens were dipped into the as-prepared boehmite-sol and were held for 20 s. The specimens were then withdrawn with a speed of 170 mm/min to produce a homogeneous film on the surface, followed by drying in an oven at 120 °C for 5 min. To assist the phase transformation from boehmite sol to alumina, a heat treatment was applied by heating the specimens inside a muffle furnace with a heating rate of 10 °C/min to 600 °C. After 2 h, the furnace was switched off to cool down the specimens.

As the first step, different coating thicknesses were used in this study, which were achieved by repeating the dipping and heat treatments two and three times, as illustrated in Figure 1. For exposure tests, the specimens were rotated alternatingly at each additional dipping step to avoid thickness differences that may occur at the top and bottom of the specimens.

For electrochemical tests, three-layer coatings were used as working electrodes to represent the Al$_2$O$_3$ coated steel. Here, the specimens were connected to Ni/Cr rods at the top of the specimens for wiring purposes and dipped in the sol entirely up to some portion of the Ni/Cr rod. A similar drying and heating procedure was performed on the specimens for electrochemical tests in which drying at 120 °C for 5 min and heating at 600 °C were carried out in between each dipping.

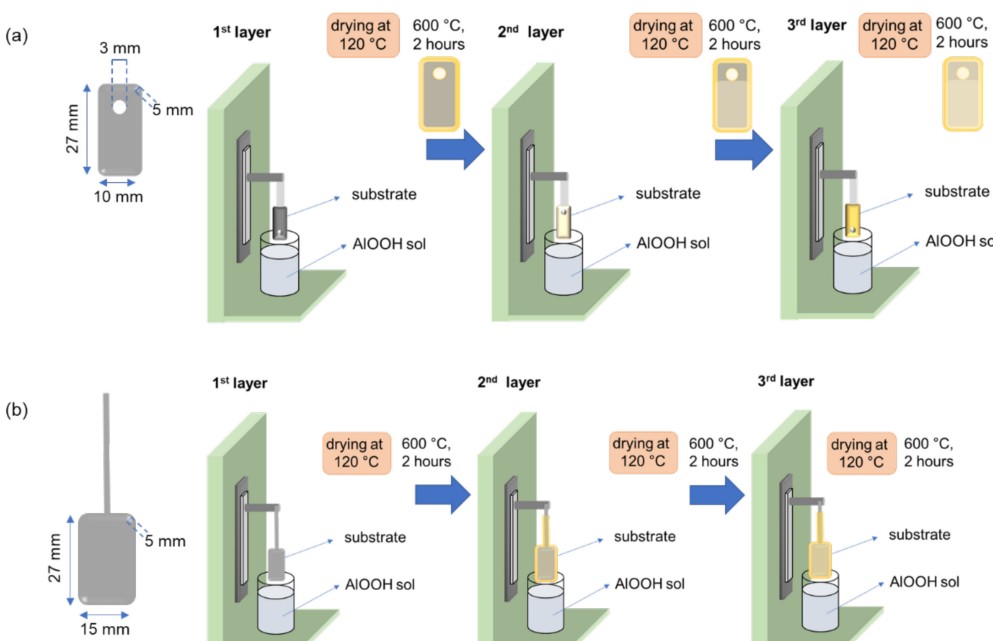

**Figure 1.** Specimen dimension and schematic procedure of $Al_2O_3$ coating preparation using sol-gel and dipping technique for (**a**) exposure tests and (**b**) electrochemical tests.

### 2.2. Exposure Test

Exposure tests were conducted to observe the corrosion of coated and uncoated steel in the artificial geothermal water. For each experiment, four identical specimens were placed in a 1-L autoclave (Parr Instrument, Moline, IL, USA) containing 500 mL of Argon-saturated artificial geothermal water. Prior to each experiment, Argon was purged for at least 30 min and additional Argon was purged through the head of the autoclaves to avoid oxygen contamination. The autoclave was equipped with a glass vessel to keep the testing solution and avoid contaminations. An initial pressure of 500 kPa was applied to each autoclave to achieve 1 MPa at 150 °C.

An artificial geothermal solution was prepared at room temperature in the laboratory, based on the chemical composition of the North German Basin (NGB) [5,32], consisting of 166 g/L $Cl^-$, 0.05 g/L $SO_4^{2-}$, 56.5 g/L $Ca^{2+}$, 0.5 g/L $Mg^{2+}$, 3.1 g/L $K^+$, 38.7 g/L $Na^+$, 0.2 g/L $NH_4^+$, and 1.55 g/L $Sr^{2+}$. The NGB solution has a pH of 6 with a specific conductivity of 219 mS/cm.

The first part of the exposure tests was used to observe the effect of coating thickness on the corrosion protection of coatings. To identify the effect of coating thicknesses on the corrosion protection, one-, two-, and three-layer coatings were exposed to the NGB solution for 7 days at 150 °C and 1 MPa. The masses of specimens were determined before and after experiments, and photos of surfaces were taken to evaluate the specimens visually.

Secondly, long-term exposure tests were performed to investigate the suitability and stability of the coating within 3 months of exposure to the artificial geothermal water at 150 °C and 1 MPa. The three-month exposure tests were performed with and without interruption. Specimens were exposed for 3 months with an interruption after 1 day, 1 week, 4 weeks, and 12 weeks. Four identical specimens were used, in which one specimen was taken out after each exposure interval for analysis. A separate experiment was performed for 3 months, in which no interruption took place. Before and after each experiment, pH and solution conductivity were measured to observe the change caused by the reactions between the specimens and the medium.



### 2.3. Characterization

The laser scanning microscope (LSM) Olympus LEXT 4100 served to investigate the surface of coated and uncoated specimens before and after exposure tests. To further analyze the changes in the surface morphology of the specimens, a scanning electron microscopy (SEM; Leo Gemini 1530VP, acceleration voltage 15 kV, Thornwood, NY, USA) was used, combined with energy-dispersive X-ray spectroscopy (EDX; EDX system Bruker, XFlash detector 5030, software Esprit 1.9, Billerica, MA, USA). On some coated surfaces, a low vacuum mode was used with a pressure of up to 10 Pa in the vacuum chamber. Cross sections were prepared by embedding the specimens in epoxy resin. These specimens were then cut, ground, and polished. Carbon sputtering was used to increase the electronic conductivity of the cross-section samples before analyzing using SEM-EDX.

### 2.4. Electrochemical Test

Short-term electrochemical tests were performed in autoclaves with an Ag/AgCl reference electrode for high temperature, Ti/TiO$_2$ counter electrode, and coated and uncoated specimens as working electrodes (Figure 2). For electrochemical measurement, three-layer coatings were used instead of one- and two-layer. The preparation step of this working electrode is shown in Figure 1. Autoclaves were heated up from room temperature to 150 °C and 1 MPa, which took 1 h to achieve. Open circuit potential (OCP) was recorded throughout the measurement to observe whether the specimens were actively corroded and to ensure that the measurement conditions were stable. EIS was recorded every hour until 24 h, with peak-to-peak sinusoidal voltage (V$_{rms}$) of 10 mV vs. OCP, within a frequency range of $10^4 - 10^{-2}$ Hz, and 8 points/decade. Gamry Echem Analyst was used to analyze the electrochemical data.

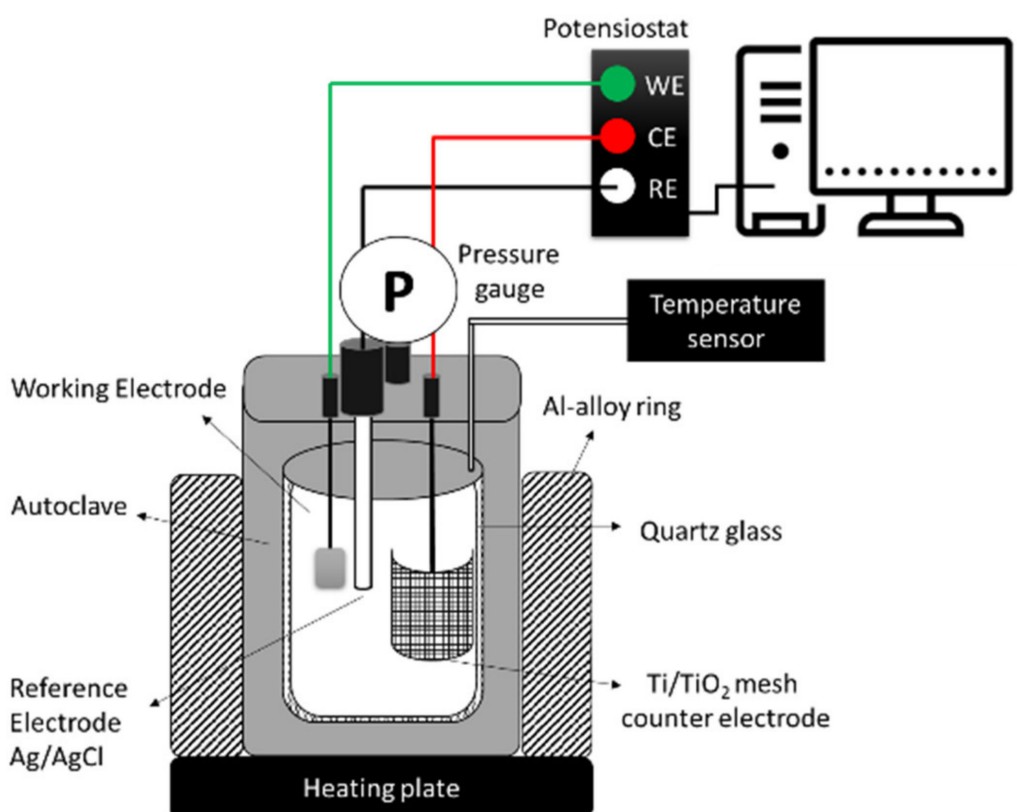

**Figure 2.** Electrochemical measurement setup.

## 3. Results

### 3.1. Effect of Coating Thickness

Alumina coatings, prepared using the sol-gel method, have thicknesses between 10 nm to 10 μm depending on the composition of the sol, conditions of deposition, and the applied heat treatment [33]. Based on previous research using similar sol formulations deposited on P92 steels and heat treated at 650 °C, the alumina coating thickness formed on the steel was approximately 1 μm [34]. To have a uniform coating surface, the average surface roughness ($R_a$) and sink height ($R_z$) of the substrate have to be lower than the estimated coating thickness. Hence, the steels were ground with P1200 to avoid the sol flow through the valleys of grinding lines and optimize the maximum coverage of coatings.

As described in the methodology section, the variation of coating thicknesses in this study was achieved by dipping the uncoated steel one, two, and three times. On the right corner of Figure 3, images of the specimens are shown, indicating that the area close to the hole is mostly delaminated, regardless of the number of layers. Such delamination may occur due to the crevice between the PTFE and the specimens which was used to place the specimens in the autoclaves. Therefore, in this study, the coating area of interest is the middle area, where there was no effect of crevice corrosion. The images of specimens after exposure tests also indicated that the one-layer and two-layer coatings underwent strong delamination. The images of the delaminated one-layer and two-layer coatings after a seven-day exposure test in the geothermal water were analyzed, resulting in approximately 50–65% and 53–79% surface coverage, respectively. Although there was a delamination of coatings, there was no indication of corrosion on the steels. To further analyze the transformation of coating morphologies before and after exposure to the NGB solution at 150 °C and 1 MPa, the unaffected areas of coatings are presented in Figure 3.

Besides the changes in the macroscopic level, microscopic structures of coatings were also altered after exposure to the NGB solution for 7 days. At the initial condition, the surface morphologies of coatings were amorphous (Figure 3a,c,e). These surface morphologies were identical for one-, two-, and three-layer coatings. Previous research on alumina sol-gel coating deposited on P92 steels has indicated that the alumina coatings with similar sol formulation and heat treated at 650 °C formed worm-like features when observed using TEM at a high magnification [24]. Based on their study, it is not possible to find a clear distinction between gamma-, delta-, and theta-$Al_2O_3$ using electron diffraction, because of the small size of the nano-crystallites stacked in the whole thickness of the TEM samples. It was also mentioned that the alumina coatings consist of connected single particles with dimensions around 5 nm, and amorphous or not fully crystalline material is situated between them. Considering that the temperature used here is 600 °C, it is suggested that at the initial condition, the alumina coatings have an amorphous or incomplete crystalline structure.

After 7 days of exposure to the NGB solution, a needle-like structure was formed on the surface of specimens (Figure 3b,d,f). Such structures are similar to the AlOOH structure reported by Serizawa et. al., in which Al-based alloys were subjected to the ultrapure water steam at 160–180 °C [35].

Cross-section images of the specimens in pristine condition show a homogeneous and continuous coverage of coatings on the steels' surfaces (Figure 4a,c,e). Based on these cross-section images, it is possible to estimate the coating thickness at the middle area of the specimen. By measuring six lines perpendicular to the cross-section images (with magnification 2000×, 150 μm length), the estimated average thickness of one layer is $1.4 \pm 0.1$ μm, the two-layer coating is $3.6 \pm 0.3$ μm, and three-layer coating is $8.4 \pm 0.3$ μm. However, it is also important to note that these local measurements do not represent the coating thicknesses at the edges, which might be highly influenced by the effect of gravitation and surface tension when the specimens were dipped and pulled out of the sol.

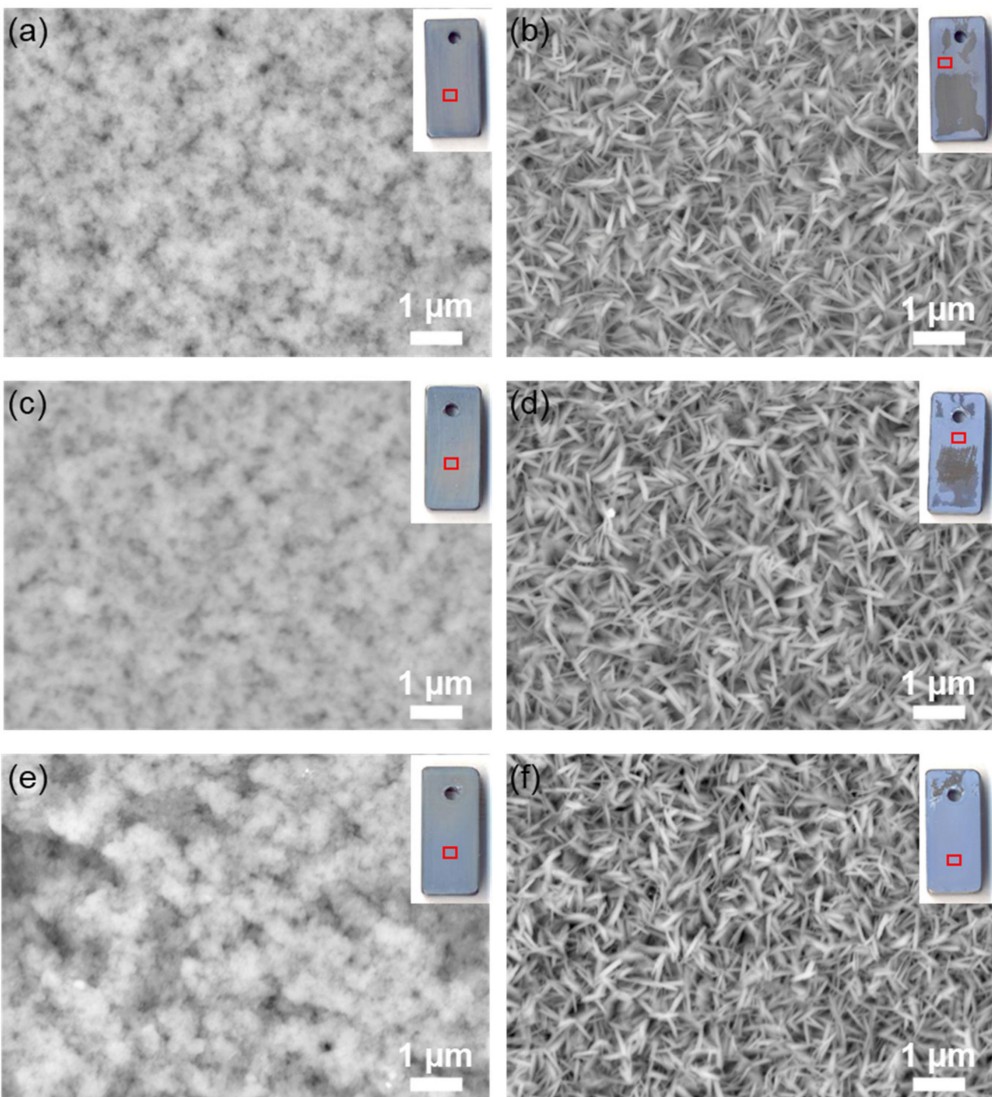

**Figure 3.** SEM images of unaffected Al$_2$O$_3$ coating morphology before (**a**,**c**,**e**) and after (**b**,**d**,**f**) being exposed to NGB for 7 days. The coatings were prepared with one layer (**a**,**b**), two layers (**c**,**d**), and three layers (**e**,**f**). Photos of each specimen are on the right corner of each SEM image.

After being exposed to the NGB solution, there were clear separations in each layer, as indicated by Figure 4b,d,f. As shown in Figure 3b,d, delamination took place in the specimens with one- and one-layer coatings; however, there was no indication of steel corrosion in the delaminated area observed in Figure 4b,d. These results indicate that the delamination of coatings observed at the macroscopic level was not influenced by the corrosion of steels, but might be influenced by the altered coating morphologies resulting from the exposure tests, leading to the lack of adhesion between each layer, and between the layer and substrate.

Here, it is suggested that there were some boehmite agglomerates built up on the surfaces of the coatings after the dipping step which are unavoidable. Such agglomerates cannot be eliminated entirely after the ultrasound treatment, as there is compensation between the stability of the sol and the viscosity [24,36]. These agglomerates might contribute to the delamination of the coating at a later stage. To have uniform coating surfaces, it is important to have sufficient coating thickness. Therefore, further experiments were performed using three-layer coatings to ensure that the steel is fully covered by the coatings.

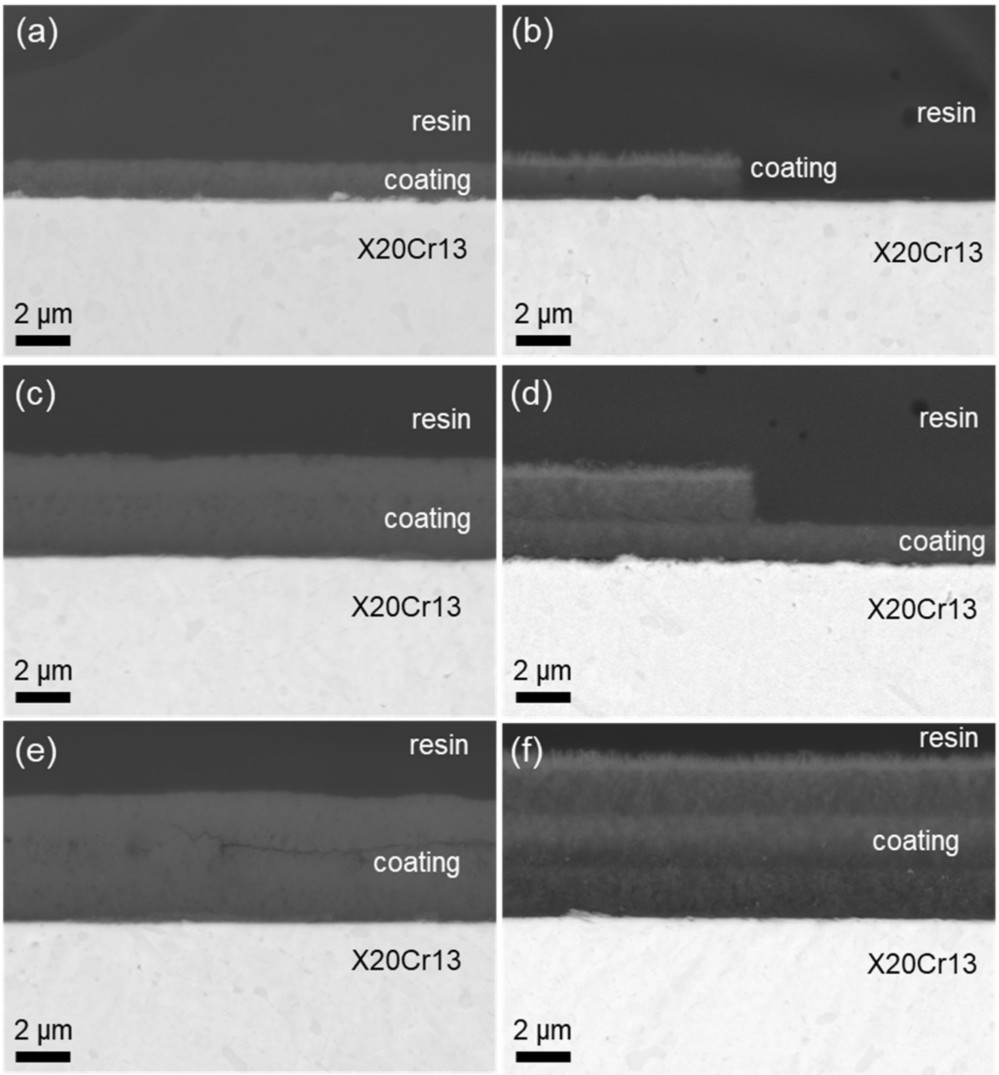

**Figure 4.** SEM images of Al$_2$O$_3$ coating cross sections before (**a,c,e**) and after (**b,d,f**) exposed to NGB for 7 days. The coatings were prepared with one layer (**a,b**), two layers (**c,d**), and three layers (**e,f**).

### 3.2. Stability of Coated and Uncoated Steels for 3 Months

To evaluate the corrosion behaviors of coated and uncoated steel, exposure tests were performed on bare and Al$_2$O$_3$-coated X20Cr13 steel with different durations up to 3 months in the NGB solution at 150 °C.

Figure 5 shows the microscopic images of uncoated X20Cr13 before and after exposure tests with different periods of time. Figure 5 shows the different colors of surfaces that might be associated with different corrosion products and/or thicknesses of corrosion product layers resulting from different exposure times. It is also possible that metastable phases of the resulting oxides and hydroxides were formed on the surface, which became more homogeneous after long-term exposure.

Figure 6 shows the microscopic images of the coated steel surfaces. At the initial condition, the grinding lines were visible, indicating the transparent feature of the Al$_2$O$_3$ coating. There were also some agglomerates observed in Figure 6a, which can be associated with the instability of the boehmite sol. This might be caused by the aggregation of boehmite particles prior to the heat treatment, which was also observed in other works [24,36]. Such unavoidable agglomerations may affect the properties and characteristics of the coating after heat treatment. Therefore, the use of three-layer coatings in this study might optimize the coating protection by reducing the effects of these agglomerates.

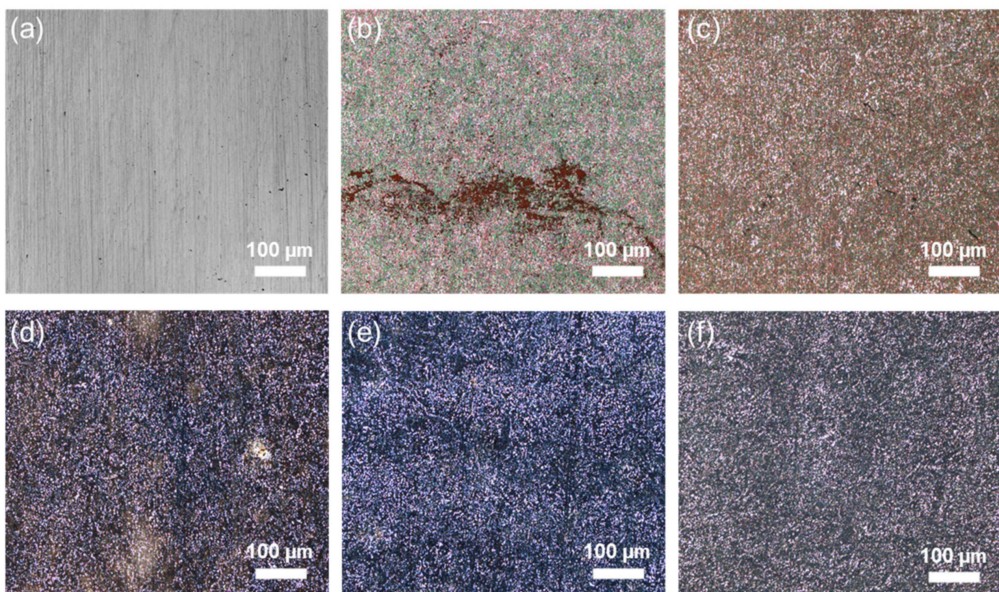

**Figure 5.** Microscopic images of uncoated X20Cr13 after being exposed to geothermal water at 150 °C, 1 MPa (**a**) before exposure. (**b**) 1 day, (**c**) 1 week, (**d**) 1 month, (**e**) 3 months, (**f**) 3 months without interruption.

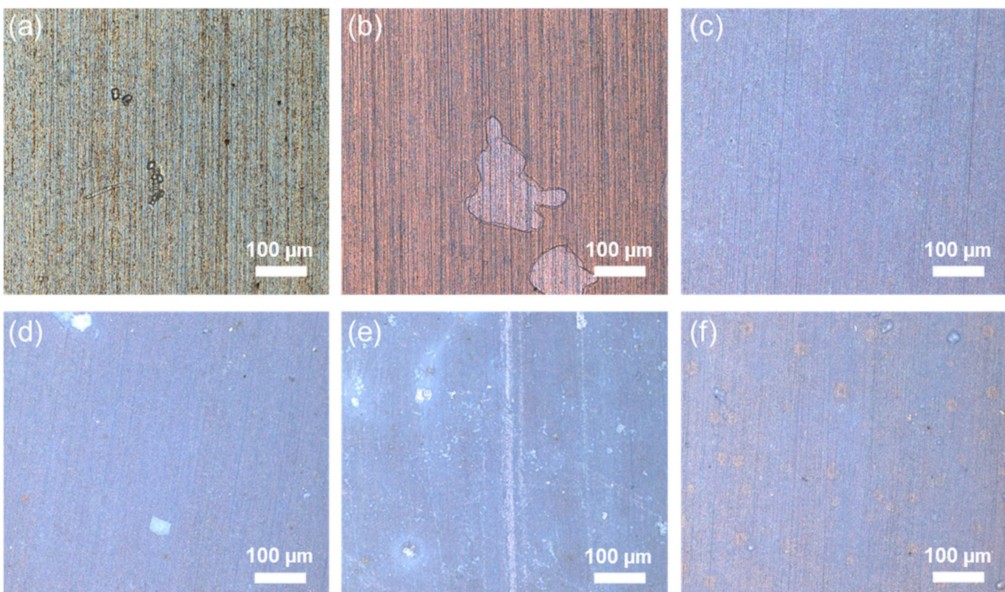

**Figure 6.** Microscopic images of three-layer $Al_2O_3$ coated X20Cr13 after exposure to geothermal water at 150 °C, 1 MPa (**a**) before exposure, (**b**) 1 day, (**c**) 1 week, (**d**) 1 month, (**e**) 3 months, (**f**) 3 months without interruption.

After being exposed to the NGB solution for one day, the coating surface was transparent and the grinding lines were still visible (Figure 6b). Although there was no significant change of color, some delaminated areas were observed. After 1 week of exposure, the grinding lines were less visible, and there was a change of coating color. This change might be associated with the evolution of surface morphology during the exposure tests. Between 1 week and 3 months of exposure, microscopic images show that there was no significant change, which might indicate that the chemical reactions between coatings and the geothermal water slowed down after the first week. Figure 6f shows that there were some heterogeneities of the coating surface, which might originate from the inhomogeneous formation and growth of AlOOH during the three-month exposure.

The solution pH was measured after each of the exposure tests (Figure 7a), in which the pH of the reacted solution in contact with uncoated specimens increased from 6 to 7 after 1 week of exposure. This pH decreased to 6.3 after 1 month of exposure, slightly lower than the solution after being in contact with the coated specimens. The pH of the reacted solution of coated specimens increased to 6.5 after 7 days, which then slightly decreased to 6.3 and remained stable until 3 months of exposure. These pH changes indicate that the most significant reaction took place in the first 7 days. Figure 7b shows that there was no significant difference in the pH measurement of the interrupted and uninterrupted three-month exposure tests.

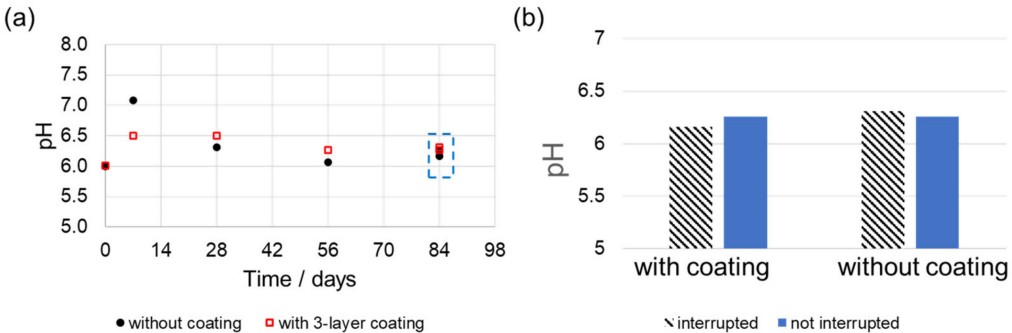

**Figure 7.** pH of solution after exposure tests (**a**) at different exposure times (**b**) after 3 months with and without interruption.

To reveal the evolution of the surface morphology over different exposure times, the exposed three-layer $Al_2O_3$ coatings were analyzed using SEM (Figure 8). After 1 day of exposure, thin needle-like structures were observed, indicating the transformation of $\gamma$-$Al_2O_3$ to AlOOH, as also mentioned in the previous section (Figure 3). As the specimens were exposed to the NGB solution for a longer time, AlOOH crystals grew into thicker needle-like structures (Figure 8b). After 1 week of exposure, there was no significant change in the AlOOH structure, which is also in agreement with the microscopic images shown in Figure 6.

Exposure tests were performed for 3 months with and without interruption to investigate whether the heating and cooling of autoclaves affect the stability of coatings. Figure 8d shows the SEM image of the coating surface after being exposed for 3 months, interrupted after 1 day, 1 week, and 1 month of exposure. There were some agglomerations identified by EDX as Al and O. The uninterrupted exposure test resulted in a more homogeneous structure, similar to that observed after 1 month of exposure. Based on this experiment, $Al_2O_3$ coatings were proven to be stable even when heated and cooled several times in the NGB solution.

To further evaluate the stability of the coatings, cross sections of uncoated and coated steel were prepared and analyzed using SEM-EDX (Figure 9). After 3 months of exposure without any interruption, steel components were dissolved due to corrosion reactions with the artificial geothermal water, and oxide layers were formed on the surface of uncoated X20Cr13 (Figure 9a). This layer consists of Fe, Cr, and O, which are also commonly formed on martensitic steels 13Cr in various corrosive environments [37]. Depending on the chemical composition, gas composition, temperature, and pressure of the corrosive medium, Fe- and Cr-oxide/hydroxide layers may be formed with different properties. Figure 9a shows that the interface between the oxide layer and steel was uneven, indicating that there might have been a competition between Fe dissolution at the beginning of the exposure test, and the formation of the oxide layer.

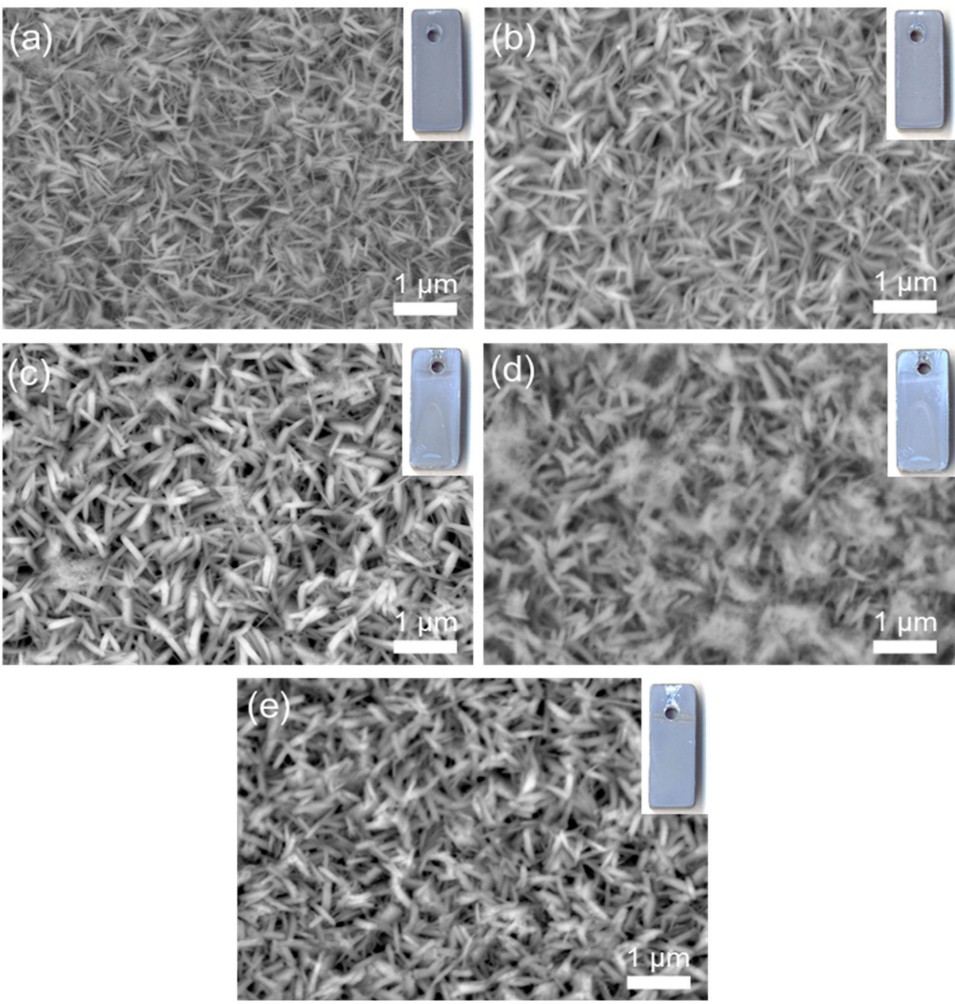

**Figure 8.** Surface morphology of three-layer $Al_2O_3$ coating after being exposed to NGB solutions for (**a**) 1 day, (**b**) 1 week, (**c**) 1 month, (**d**) 3 months, and (**e**) 3 months without interruption.

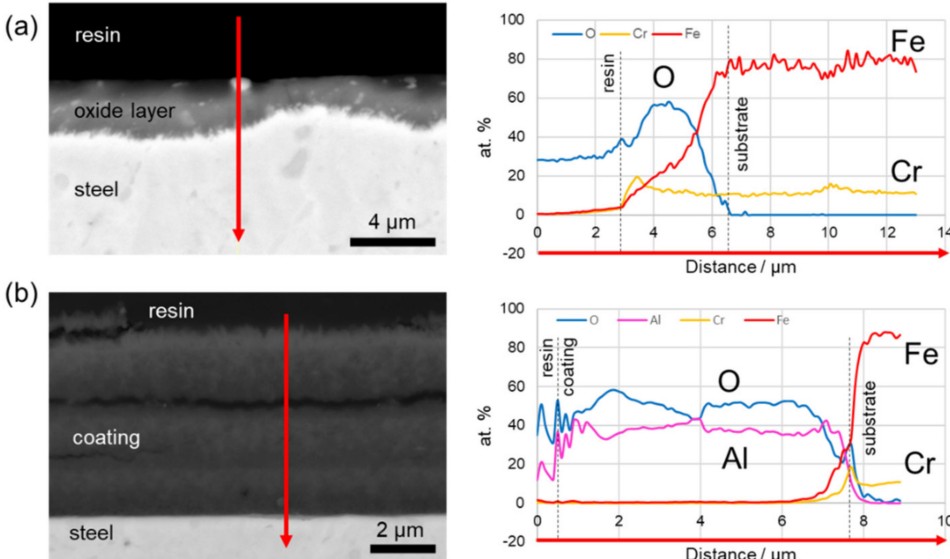

**Figure 9.** SEM images and EDX of cross section for (**a**) uncoated X20Cr13 and (**b**) $Al_2O_3$-coated X20Cr13; after 3 months in the NGB at 150 °C, 1 MPa without interruption.

Figure 9b shows that the coated area consisted of Al and O, showing that the boehmite layer formed at the beginning of exposure is protective in the NGB solution during three-month exposure. The transformation of the boehmite layer took place on the most outer part of the coating with an approximate depth of 1 μm. This can be estimated based on the point where the EDX spectrum shows an increase of oxygen after 1 μm depth. Similar to Figure 4f, each layer of the coatings was distinctive, which might lead to poor adhesion between each layer. Although there was a clear separation between each layer, coatings were able to protect the steel substrate, without any indication of pitting corrosion or uniform corrosion and no loss of the steel surface. A thin Cr-enriched zone was observed at the interface between the coating and substrate (Figure 9b), which could contribute to the corrosion protection of steels. This result indicated that the chromium within the steels could assist with the corrosion protection at spots where coating imperfections existed. Thus, the three-layer coatings could effectively act as chemical and physical protection barriers on the X20Cr13 steels.

### 3.3. Electrochemical Behavior of Coated and Uncoated Steels

To understand the early stage of interaction between the $Al_2O_3$ coatings and the NGB solution, in-situ electrochemical impedance measurements were performed for both three-layer-coated and uncoated steel. Figure 10 shows the absolute impedance values at 0.01 Hz ($|Z|_{0.01Hz}$) with respect to time. Throughout the 24-h measurement, $Al_2O_3$-coated X20Cr13 steel has much higher $|Z|_{0.01Hz}$ compared to the uncoated X20Cr13. As shown in Figure 10, $|Z|_{0.01Hz}$ of coated X20Cr13 was approximately 10 times higher at the first hour and gradually decreased and stabilized after about 5–6 h of immersion in the NGB solution. This significant decrease of absolute impedance value is commonly observed in coatings and can be associated with the water uptake within the coatings. The water uptake will affect the coating capacitance, leading to the dielectric constant change of the reacted coatings. The impedance spectra are discussed in detail to observe the changes in uncoated X20Cr13 steels (Figure 11) and three-layer-$Al_2O_3$-coated steels (Figure 12) after 1, 3, 6, 12, 18, and 24 h of immersion.

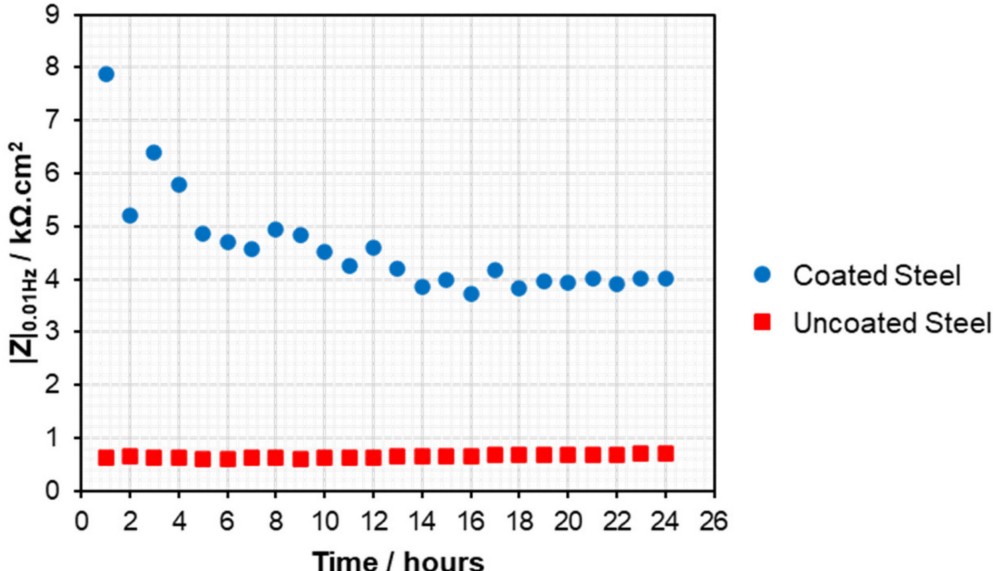

**Figure 10.** Comparison of absolute impedance value at 0.01 Hz of X20Cr13 with and without $Al_2O_3$ coatings in NGB solution at 150 °C.

Figure 11a shows that there was an increase of absolute impedance $|Z|$ along with exposure time, which can be associated with modification of the passive layer, either as the formation of different film compositions or growth of a thicker film on the surface. As shown by Figure 11b, two time constants were identified, with relatively similar phase angle

peaks at $-56°$ in the mid-range frequency and at about $-27°$ at the lower frequency. There was not much change in impedance spectra during the 24 h of measurement, indicating very slow growth of the passive layer.

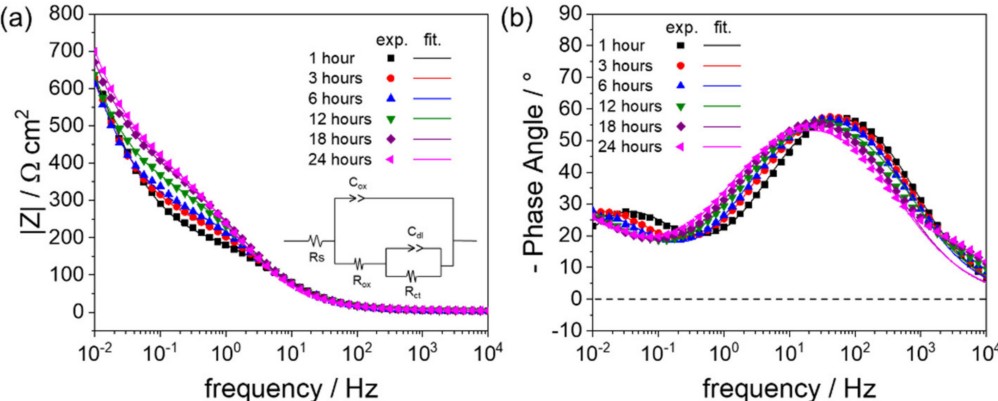

**Figure 11.** Impedance spectra of uncoated X20Cr13 in NGB solution at 150 °C (**a**) Bode plot, (**b**) Phase angle plot.

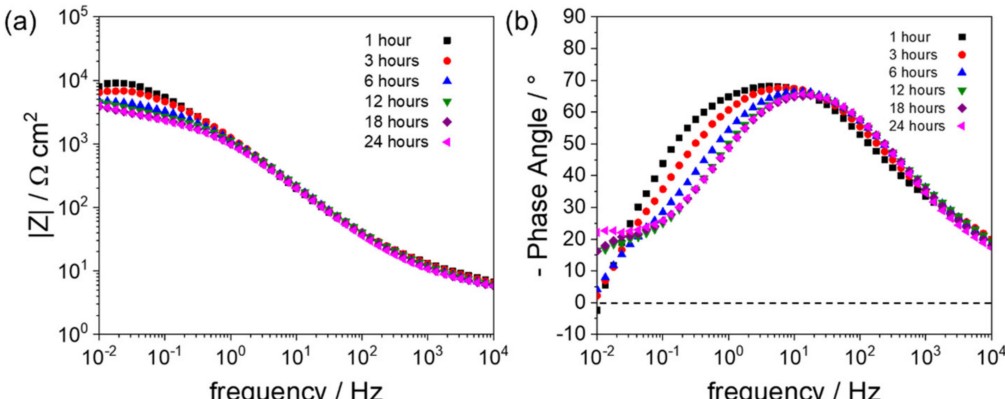

**Figure 12.** Impedance spectra of Al$_2$O$_3$-coated X20Cr13 in NGB solution at 150 °C (**a**) Bode Plot, (**b**) Phase Angle Plot.

Marcelin et al. discussed the impedance spectra of martensitic steel X12CrNiMoV12-3 in a neutral chloride solution 0.1 M NaCl + 0.04 M Na$_2$SO$_4$, it showed very high absolute impedance in the order of almost $10^6$ $\Omega$ cm$^2$ after 17 h of measurements in both aerated and deaerated conditions [37]. The electrochemical impedance analysis provided an insight into the short-term change in the passive layer formed on martensitic steels at room temperature. However, as shown in Figure 11, in-situ EIS measurement of martensitic steels at 150 °C and 1 MPa resulted in a maximum absolute impedance of almost $10^3$ $\Omega$ cm$^2$ after 24 h in the solution, which shows that the corrosion resistance of martensitic steel is much lower when exposed to a higher temperature and pressurized aqueous system compared to those at the room temperature.

In contrast with the uncoated steel, $|Z|$ of Al$_2$O$_3$-coated steel decreases over time, which can be clearly identified at the lower frequency range (Figure 12a). The phase angle plot shows that there was an additional time constant after 6 h of immersion in the solution. This might be associated with the water uptake at the beginning of immersion, followed by the transformation of Al$_2$O$_3$ to AlOOH, as observed in the SEM images of the specimens before and after exposure tests, indicating the change in surface morphology from that demonstrated in Figure 3e (at the initial condition) to the one in Figure 8a (after 24 h). Regardless of the immersion time, the phase angle at the mid-frequency region was stable at about $-60°$ to $-65°$, which was lower than that of uncoated steels,

indicating more capacitive behavior of coatings. As revealed by the cross-sectional images in Figures 4 and 9, $Al_2O_3$ layers were transformed and each layer was partially separated, which might allow additional pathways for the electrolyte to penetrate through the coatings. Such a mechanism leads to quite complex equivalent circuit models that should be first rationalized by experiments in a simpler system.

## 4. Discussion

### 4.1. Transformation of $Al_2O_3$ to AlOOH and Its Stability

As previously mentioned, $\gamma$-$Al_2O_3$ coatings have been successfully used in different corrosive environments. In a high-temperature dry atmosphere, $Al_2O_3$ coatings remained intact and protective even after 1000 h of exposure, showing the excellent properties of the coatings [27]. However, studies on the use of $\gamma$-$Al_2O_3$ coatings for corrosion protection in a water-based solution at high temperatures are limited, due to the possible rehydration of $Al_2O_3$. Carrier et al. mentioned that $\gamma$-$Al_2O_3$ tends to form boehmite AlOOH when in contact with a water-based solution at pH lower than 5, whereas bayerite or gibbsite $Al(OH)_3$ are more favored to form in the solution with higher pH [38]. The formation mechanism of these polymorphs strongly depends on the experimental conditions, which include pH, temperature, pressure, and the nature of ions in the solution [38].

Depending on the chemical composition and pH of the solution, the stability of AlOOH or $Al(OH)_3$ layers formed on the surface might also varies. Hence, they could also be protective in a corrosive environment. For example, Ruhi et. al. investigated coated and uncoated AISI 304L in 0.01 M $H_2SO_4$ containing 0.01 M, 0.025 M, and 0.05 M NaCl at room temperature [39]. They reported that the coated steels have better corrosion resistance than the uncoated steels. In addition, at the same pH, the different chloride concentrations influence the kinetics of corrosion processes. Based on these studies, $Al_2O_3$ sol-gel coating has the potential to be used as a protective coating in aqueous systems, as well as in geothermal applications.

In the NGB solution, the amorphous $\gamma$-$Al_2O_3$ transformed into crystalline needle-like structures within one day of exposure. A similar structure as shown in Figure 3 was also observed elsewhere, indicating the formation of AlOOH crystals [35]. In their study, Serizawa et. al. reported that AlOOH was formed on the surface of an Al-based alloy after exposure to ultrapure water steam at 160–180 °C for 1 h [35]. The corrosion resistance of the AlOOH-coated Al-based alloy was also measured in 5 wt% NaCl solution at room temperature, which showed that the AlOOH-coated Al-based alloy had a higher corrosion resistance than the uncoated Al-based alloy. In addition, there was no pitting corrosion even after 48 h of immersion. Notably, this study has similar testing conditions to our study, because it was carried out in autoclaves within 160 °C–180 °C, although with lower pressure between 0.6–0.95 MPa. Therefore, it is possible that regardless of the complex chemical composition of the NGB solution, AlOOH was first formed due to the contact of $Al_2O_3$ with water.

As presented by the EIS results of $Al_2O_3$ coatings, a significant decrease of absolute impedance value was observed in the first 6 h (Figure 10), which can be associated with the water uptake within the coating, followed by a transformation from $Al_2O_3$ to AlOOH. Thin needle-like structures were observed after 1 day of exposure, which confirms a fast reaction between the coatings and the NGB solution (Figure 8a). After being exposed to NGB for 1 week, thicker and denser crystalline AlOOH was formed and remained stable for 3 months of exposure. At the same time, the pH of the solution increased from 6 to 6.5 within 1 week and remained stable until 1 month of exposure. A slight decrease from 6.5 to 6.3 was observed after 3 months of exposure. This further confirms that the reaction between the coatings and the solution slowed down after the first week of exposure, and AlOOH was relatively stable until 3 months of exposure.

*4.2. Al₂O₃ Coatings on X20Cr13 Steels for Geothermal Application*

In this study, martensitic steel X20Cr13 was used as a substrate because it is the most affordable Cr-containing steel, often applied as tubing material in both oil and gas, and geothermal environments [40]. Due to their ability to form a passive layer, martensitic 13Cr steels have slightly better corrosion resistance than carbon steel [41]. However, several studies also showed that these steels might be suffering from pitting corrosion. Pfennig et al. investigated 13Cr steels including X20Cr13 and X46Cr13 in the Stuttgart aquifer artificial geothermal solution. After being immersed for 6000−8000 h in the $CO_2$-saturated aquifer water at 60 °C, X20Cr13 sounded more pitting corrosion than other steels [32,42]. Yevtushenko et al. also reported that X46Cr13 underwent localized corrosion in Stuttgart aquifer geothermal water at 60 °C, with and without $CO_2$ [7].

Due to the demand for pipeline materials for high-temperature applications, corrosion studies of 13Cr steels have also been performed at higher temperatures, for example, up to 250 °C [40,43]. However, there is still a lack of understanding of the corrosion mechanism of 13Cr steels because of the complex corrosion processes, especially the formation of the passive layer and its protectability in the medium. For example, 13Cr steels have been tested in Br-containing aerated artificial brine at 150 °C [44]. The study showed that more pitting corrosion is expected at higher temperatures. These studies suggest that 13Cr-steels are susceptible to corrosion in several geothermal environments, which necessitate their protection from coatings or inhibitors.

As shown by the morphological characterization, thin oxide layers were formed on the surfaces of the steels (Figure 5). An exposure test for 3 months confirmed that the layer formed on the steel surface consists of Fe, Cr, and O as the most dominating elements, which might be associated with (Fe,Cr) oxides and hydroxides. Marcelin et. al. reported that in deaerated neutral chloride solution, (Fe,Cr) oxides and hydroxides were slightly changed even at room temperature [37]. Although there was no significant corrosion, their study pointed out the possible evolution of the passive layer even in a less aggressive corrosive environment. Meanwhile, at a higher temperature and pressure, the high ionic activities might also influence the formation of the corrosion products. Regardless, impedance spectra showed that uncoated X20Cr13 has lower impedance value ($<10^3$ $\Omega$ cm$^2$) than $Al_2O_3$ coated X20Cr13 ($<10^4$ $\Omega$ cm$^2$), indicating an enhancement of corrosion protection provided by the $Al_2O_3$ coatings and a thin chromium enriched layer on the steel surfaces of the coated samples.

Impedance spectroscopy is a useful technique to identify the change in overall impedance of the specimens over time. However, these measurements cannot distinguish the surface from species located close to the surface. Consequently, to fit a meaningful equivalent electrical circuit model on this condition is quite complex. Despite these limitations, EIS was able to show the crucial immersion time, in which the significant changes of coatings took place and became more stable. Therefore, future work should focus on the analysis of impedance spectra measured at high temperature with a wider frequency range and longer immersion time to extract more information regarding the corrosion protection mechanism of the alumina coatings.

## 5. Conclusions

Thin $Al_2O_3$ coatings were applied on X20Cr13 to protect steels in artificial geothermal water, that is, the NGB solution at 150 °C and 1 MPa. An optimization step was evaluated by using different thicknesses of coating. Short-term exposure tests of one-, two-, and three-layer coatings performed at 150 °C and 1 MPa, resulting in the delamination of the one- and two-layer coatings. Therefore, three-layer coatings were used, as they provided the best protection on X20Cr13. Based on the morphological characterization, the $Al_2O_3$ coating was transformed into crystalline AlOOH after being exposed to the NGB solution, regardless of the number of layers applied.

The transformation of $Al_2O_3$ into AlOOH was identified after only 1 day of exposure where thin needle-like structures were observed by SEM. Further exposure tests resulted

in the growth of AlOOH crystals to bigger and denser structures. The crystalline AlOOH was stable until up to 3 months of exposure to NGB solution at 150 °C, 1 MPa. In addition, EDX analysis of the $Al_2O_3$-coated steel showed that there was no Fe and Cr detected on the surface of coatings after 3 months of exposure, which suggests that the coating was protective. In contrast, the uncoated X20Cr13 steels were corroded upon exposure to the NGB solution at 150 °C, 1 MPa, forming thin oxide layers right from the first day of exposure.

Short-term electrochemical measurements using EIS revealed that coatings have an impedance of 10 times higher at the first hour. These values significantly decreased to 5–6 times after 6 h and became relatively stable until 24 h. This result confirms the fast reactions of $Al_2O_3$ with the artificial geothermal solution. In comparison to the uncoated steels, the absolute impedance value remained higher, suggesting a better corrosion resistance. In the future, EIS can be further used for a longer immersion time and a wider frequency range to investigate the coating protection mechanism in detail.

Lastly, there are several important criteria of coatings that have to be considered before using the coatings in geothermal applications. These criteria include mechanical abrasion, compatibility in the coefficients of thermal expansion (CTE) between matrix and filler, adhesion of coatings to substrates, and ease to repair [9]. Here, $Al_2O_3$ coatings synthesized using the sol-gel method were proven to be thermally stable in geothermal water at 150 °C and 1 MPa and contributed to the increased corrosion resistance. However, this coating should be further developed and investigated to evaluate the effect of crevice corrosion, along with adhesion and mechanical properties of coatings before utilizing them in real geothermal pipelines.

**Author Contributions:** Conceptualization, G.A., M.N. and R.B.; methodology, G.A., M.N., R.S. and R.B.; validation, M.N. and R.B.; formal analysis, G.A.; investigation, G.A.; resources, R.B.; data curation, G.A., L.Q.H. and M.N.; writing—original draft preparation, G.A.; writing—review and editing, L.Q.H., M.N., R.S. and R.B.; visualization, G.A.; supervision, R.B.; project administration, R.B.; funding acquisition, R.B. All authors have read and agreed to the published version of the manuscript.

**Funding:** This research was funded by BAM, the Federal Institute for Materials Research and Testing.

**Institutional Review Board Statement:** Not applicable.

**Informed Consent Statement:** Not applicable.

**Data Availability Statement:** Data Availability is given according to DFG´s Code of Conduct "Safeguarding Good Research Practice" and complies with "MDPI Research Data Policies" at https://www.mdpi.com/ethics.

**Acknowledgments:** The authors would like to thank H. Strehlau for the preparation of geothermal solutions and elemental analysis of the X20Cr13 initial composition, R. Saliwan-Neumann for the assistance in taking SEM images, and S. Engel for the help in preparing cross-sectional samples.

**Conflicts of Interest:** The authors declare no conflict of interest.

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
