# Peer review of "Study of Al2O3 Sol-Gel Coatings on X20Cr13 in Artificial North German Basin Geothermal Water at 150 °C"

_coatings, doi:10.3390/coatings11050526_

Round 1

Reviewer 1 Report

In this work, the authors conducted an interesting work on the Al2O3 Sol-Gel coatings for X20Cr13 steel, and then analysed the corrosion properties under the artificial North German Basin (NGB) solution with different exposure hours at 150 °C with three different layer coatings. The results show the Al2O3 coating, especially 3-layer coatings, provided protection on X20Cr13, with surface transform to AlOOH crystals, which was stable until up to 3 months of exposure in NGB solution. The authors show a clear background of why the coating is necessary for the martensitic stainless steels, like X20Cr13 on the service conditions, especially at high temperature, and the advantage of Al2O3 coating.

I have some suggestions or questions to the authors.

  1. Page 2 line 93-95: The description is not clear. Do all exposure tests and electrochemical test specimens need to draw a hole on the specimen or not? In my opinion, all the specimens need to draw a hole for the sol-gel coating. Is that possible for the author to draw a draft for the specimen size, which would be helpful for the reader's understanding.
  2. Page 3 line 101-102: I understand why the authors wrote the roughness on experimental method section, because only one sentence could clearly describe the roughness data, but I am still thinking, it may better in the result section, especially on Page 4 line 155 in characterization section, the authors mention the LSM again for the specimen before and after exposure test. The roughness after grinding should be the condition before coating, but it also relates to the condition before the exposure test.
  3. Page 5 line 179-182: Those are the background of the Al2O3 coating. Shall the authors move to introduction?
  4. Page 6 line 212: I cannot see the delamination from Fig. 3b and 3b, but could observe this from Fig. 4b and 4d and with no delaminated in Fig. 4f.
  5. Page 7 line 247: I could still observe the grinding lines after 1 week and one month specimen. But it is difficult to observe from the specimen without coating after 1 day. Does my eyes have problems?
  6. Page 10 line 317: Al2O3-coated steel-is this 1-layer, 2 layer or 3 layer specimen?
  7. Page 11 line 343: Does the authors supply the SEM image to prove the Al2O3 transformation to ALOOH between 1to 24h? Because Fig. 7 showed the images from 1 day.
  8. Page 12 line 393-396: There is no data that shows the PH change from exposure in the result section.
  9. Page 13 line 449: I think the conclusion that the authors gave focus on the 3 layer specimen, because the author only shows the exposure data to 3 month on the 3 layer specimen.
  10. Page 13 line 454: I think the authors could be better to clarify that there is no Cr in the Al2O3 coating surface than only give the EDS data from uncoated specimens.

In conclusion, the authors did great work on the Al2O3 coatings for X20Cr13 steel, although there are still few things that need to explain clearly. But the paper is the whole story to show the importance of the Al2O3 coating on the protection of the X20Cr13 steel, although there are still few further works to do.

Author Response

Pleas see the attachment

Reviewer 2 Report

Review Coatings

Study of Al2O3 sol gel coatings on X20Cr13 in Artificial North German basin geothermal water at 150 °C

The paper presents a study on the properties of Al2O3 sol gel coatings in synthetic solution of geothermal water at very high temperature of 150 °C.

Different methods for evaluating properties of the coating were employed. All together 11 figures are used, 2 tables and the study refers to 43 references.

Abstract: ok

Keywords

Ok

Introduction. Ok

Materials and methods:

Line 92 X20Cr13 steel was used as substrate material with chemical composition given in Table 1.

Authors say that the cleaning procedure after grinding was rinsing with bi distilled water and ethanol. Are authors aware that it is essential that samples are ultrasonically cleaned in order to get rid of SiC particles before applying sol gel coating?

Line 103: Why is roughness information (with Sing height higher than roughness) important for their study?

Line 121: How was coating with 3 layers prepared? Just dipping three times without curing in between?

Line 128-129: every time when you say geothermal water use also temperature and pressure because all these information add to corrosiveness of the media.

Line136-139: it would be of good information if at least [Cl-] concentration is given also in molarity. Is it really so concentrated that 2.8 mol/L of Cl- is present?

Line 173:

EIS measurements: why did authors choose only 10 mHz as the lowest frequency, due to temperature and oscillations? So that they could perform EIS hourly?

When studying coatings, usually 1 mHz or at least 5 mHz is tried to be measured as well.

Results

Line 179: effect of coating thickness would be a better subchapter title.

When you talk about coating thickness, you define 1 layer, 2 layer and 3 layers. What are the thicknesses of 1 layer, 2 layer and 3 layer coating?

Figure 3: define the % of coverage are after exposure. Just description is not enough.

What are needle like products. Did you analyze them (Raman, XRD) or at least find similar data in literature?

From figure 6- a kind of pitting is observed. The analysis of such area would be essential. The authors should analyze the orange points in figure f in detail.

Figure 8: This is a very nice analysis, but it is done in one linear line. You have chosen that in a place, where the coating is intact, what about at the place close to a hole in figure 7e?

You should also anise the weaknesses of the coating.

Figure 9: what happens after 24 h? It seems that EIS measurement could be easily performed at least 1 weak? Did the measurements show the reduction of the impedance response once coating failed. Why not showing such results as well? It is the weakest point which will deteriorate the whole system.

Lines 257-374: the authors offer a lot of information in literature data, but, did you analyze your own coating?

Conclusions:

Would you use such coating for your tubing from X20Cr13 steel in geothermal water at 150 °C?

Reviewer 3 Report

This paper is a rigorous work and has a major contribution to improve the corrosion resistance of Cr-steels in geothermal water by obtaining protective coatings. Al2O3 it was chosen as a potential coating. The written and quality of the manuscript are acceptable. The authors reviewed all aspects related to the success of this coating method and systematic and long-term corrosion tests were carried out. Besides, the organization of this paper is also properly arranged, and easily understood, and the topic is suitable for ‘Coatings’. The reviewer recommends the article for publication after addressing the following concerns.

  1. Indicate the initial state of the steel taken for research.
  2. ‘Specimens were exposed for 3 months with an interval of … 12 weeks’. It is not clear what is meant.
  3. ‘At the initial condition, the surface morphologies of coatings were amorphous’. How was this determined?
  4. ‘After 7 days of exposure to the NGB solution, a needle-like structure was formed on the surface of specimens’. What is the depth of the coating transformation?

Reviewer 4 Report

The manuscript concerns study of Al based coating on steel in geothermal water. The manuscript is interesting and very well prepared. I have no major objections to the prepared description.

My only remark concerns: 3.1; lines 222-226; the authors write about the imperfections of the coatings. On the one hand, the imperfections may be reduced by 3-layer coatings. On the other hand, if such imperfections occurred, the sample preparation might have to be repeated in order to avoid them.

Additionally; lines 185-188, please unify the temperature ranges.

Round 2

Reviewer 2 Report

The authors have tried to answer all the risen questions and the paper is improved to an extent that can be published.